# Selection of Mexican Medicinal Plants by Identification of Potential Phytochemicals with Anti-Aging, Anti-Inflammatory, and Anti-Oxidant Properties through Network Analysis and Chemoinformatic Screening

**DOI:** 10.3390/biom13111673

**Published:** 2023-11-20

**Authors:** Oscar Salvador Barrera-Vázquez, Sergio Andrés Montenegro-Herrera, María Elena Martínez-Enríquez, Juan Luis Escobar-Ramírez, Gil Alfonso Magos-Guerrero

**Affiliations:** 1Department of Pharmacology, School of Medicine, Universidad Nacional Autónoma de México (UNAM), Mexico City 04510, Mexico; osbarrerav@comunidad.unam.mx (O.S.B.-V.); emartinez@facmed.unam.mx (M.E.M.-E.); jlescobar@comunidad.unam.mx (J.L.E.-R.); 2Department of Basic Medical Sciences, Faculty of Health Sciences, Icesi University, Cali 760031, Colombia; samontenegro@icesi.edu.co

**Keywords:** native, Mexican, medicinal, plants, anti-oxidants, anti-inflammatory, senolytic, database, anti-aging, phytochemicals, network

## Abstract

Many natural products have been acquired from plants for their helpful properties. Medicinal plants are used for treating a variety of pathologies or symptoms. The axes of many pathological processes are inflammation, oxidative stress, and senescence. This work is focused on identifying Mexican medicinal plants with potential anti-oxidant, anti-inflammatory, anti-aging, and anti-senescence effects through network analysis and chemoinformatic screening of their phytochemicals. We used computational methods to analyze drug-like phytochemicals in Mexican medicinal plants, multi-target compounds, and signaling pathways related to anti-oxidant, anti-inflammatory, anti-aging, and anti-senescence mechanisms. A total of 1373 phytochemicals are found in 1025 Mexican medicinal plants, and 148 compounds showed no harmful functionalities. These compounds displayed comparable structures with reference molecules. Based on their capacity to interact with pharmacological targets, three clusters of Mexican medicinal plants have been established. *Curatella americana*, *Ximenia americana*, *Malvastrum coromandelianum*, and *Manilkara zapota* all have anti-oxidant, anti-inflammatory, anti-aging, and anti-senescence effects. *Plumeria rubra*, *Lonchocarpus yucatanensis*, and *Salvia polystachya* contained phytochemicals with anti-oxidant, anti-inflammatory, anti-aging, and anti-senescence reported activity. *Lonchocarpus guatemalensis*, *Vallesia glabra*, *Erythrina oaxacana*, and *Erythrina sousae* have drug-like phytochemicals with potential anti-oxidant, anti-inflammatory, anti-aging, and anti-senescence effects. Between the drug-like phytochemicals, lonchocarpin, vallesine, and erysotrine exhibit potential anti-oxidant, anti-inflammatory, anti-aging, and anti-senescence effects. For the first time, we conducted an initial virtual screening of selected Mexican medicinal plants, which was subsequently confirmed in vivo, evaluating the anti-inflammatory activity of *Lonchocarpus guatemalensis* Benth in mice.

## 1. Introduction

Inflammation, oxidative stress, and cellular senescence are involved in age-related diseases, degenerative diseases, and infections [1,2,3,4].

Inflammation is a manifestation of the innate immune system’s defensive and adaptive response against injuries and harmful agents, such as bacteria, viruses, and toxins, in order to re-establish homeostasis. A dysregulated inflammatory response activates phytological processes, which can cause sepsis and organ failure. Chronic inflammation can occur because of pro-inflammatory cytokine secretion stimulated by senescent cells [5]. The phenomenon of low-grade chronic inflammation is characteristic of human aging and is termed “inflammaging” [2,6]. The para-senescence process, which is triggered by secretory phenotypes linked to senescence (SASPs), leads to the generation of persistent inflammation [7]. As a result, the inflammatory cascade causes an increase in extracellular ROS concentrations and oxidative stress. Elevated oxidative stress can cause the breakdown of the extracellular matrix (ECM) and activation of cell necrosis and apoptosis [5].

Aging and chronic degenerative pathologies exhibit common features, including elevated bioavailability of reactive oxygen species (ROS) and oxidative stress, chronic/persistent inflammation, glycation, and mitochondrial abnormalities. The production of excessive ROS leads to the destruction of nucleic acids and proteins, resulting in alterations to the cellular structure and functional outcome [5].

Oxidative stress arises from diminished anti-oxidant levels and the disturbance of the dynamic redox circuitry system, leading to the accumulation of free radicals within the body. The occurrence of oxidative stress initiates a negative sequence of events, leading to the modification of the chemical structure of cells, deterioration of the cell membrane, obstruction of crucial enzyme activities and energy production, and hindrance of vital cellular processes necessary for the body’s regular functioning [8]. Anti-oxidants have a vital function in the delay or prevention of cellular damage and the provision of essential protection against oxidative stress. Among the noteworthy enzymatic anti-oxidants are superoxide dismutase (SOD), glutathione peroxidase (GPX), catalase (CAT), and thioredoxin (Trx). The enzyme SOD facilitates the transformation of O_2_•− into the less reactive H_2_O_2_. GPX catalyzes the decomposition of H_2_O_2_ and lipid hydroperoxide (LOOH), while CAT catalyzes the oxidation of H_2_O_2_ to H_2_O and molecular oxygen (O_2_), and Trx catalyzes the conversion of H_2_O_2_ to produce H_2_O. To counteract or neutralize the effects of free radicals, the human body generates anti-oxidants. The stability of free radicals and anti-oxidants is vital for the appropriate physiological function of the body [5].

Natural Products have been used since ancient times in drug development. Between 1981 and 2019, 36.3% of the small molecules were developed for the treatment of antimicrobial, antiparasitic, and anticancer conditions based on natural products or their derivatives [9,10,11]. Less than 20% of it is sourced from plants, as stated in reference [12]. Cognitive and cardiovascular health benefits can come from consuming flavonoids, terpenoids, saponins, and polysaccharides like astragaloside, ginkgolide, ginsenoside, and gipenoside. The U.S. FDA has designated the term “Botanical Medicines” to refer to plant components that comprise complex mixtures. These mixtures require both chemical quality control and a manufacturing validation process [9].

For centuries, traditional medicine has been a fundamental part of the primary healthcare of populations worldwide [13,14]. According to anthropological and archaeological data, medicinal plants were used in traditional Mexican medicine for over 5000 years. Using medicinal plants has been passed down through oral tradition, and many of these plants continue to be used as their historical usage [15]. Mexico has 28,906 types of vascular plants that are grouped in over 2900 genera, 319 families, and 76 orders, according to EncicloVida [16]. Its number of endemic species (approximately 50%) is second only surpassed by South Africa [17]. Today, the number of plant species used in traditional medicine in Mexico exceeds 3000; the work “Atlas de las Plantas de la Medicina Traditional Mexicana” records 3103 [18], while an independent study recorded 3352 species [19]. Enciclovida records 1025 species of native medicinal plants [20] consulted in June 2023.

Medicinal plant species, either native or introduced, are used by over 52 different ethnic groups. Herbal remedies are preferred by over 90% of Mexicans for their effectiveness, affordability, and fewer side effects compared to allopathic medicine [21].

Approximately 85% of health professionals are aware of herbal medicines, and  approximately 75% recommend their use [22,23]. Traditional herbal medicine remains valuable in rural and indigenous communities despite modern healthcare. Herbal medicines are important in traditional medicine for culturally appropriate and accessible healthcare. Because of a lack of insurance and cultural beliefs, Mexican immigrants in the U.S. use alternative medicines. In Texas–Mexico border cities, 68.3% of pharmacists find patients using complementary and alternative medicines [21].

In addition, some states maintain botanical and medicinal plant gardens. Specialized stores sell traditional medicinal plants and medicines without proper quality control and regulation. In the 1990s, the Ministry of Health introduced a classification system to evaluate the safety and efficacy of herbal remedies used in traditional Mexican medicine. The NOM-072-SSA1-2012 [24] standardized and defined legal uses and packaging of herbal remedies since 2013. However, this regulatory law is not in force throughout Mexico [25].

Computational tools and novel algorithms have accelerated and optimized the drug discovery process. It is estimated at 20 years and saves around USD 1.3 billion [24]. Several methods have been shown to reduce drug development costs by up to 50%. Methods include chemoinformatics, QSAR, docking, molecular similarity, network pharmacology, and computational design [21,26]. Bioinformatic development has improved computer and web tools for designing new drugs. The analysis of compound databases using chemoinformatics has led to the discovery of new drugs.

The aim is to identify Mexican medicinal plants (MMPs) with one or more potential anti-oxidant, anti-inflammatory, anti-aging, and anti-senescence (AOX-AINF-AAG-ASEN) effects. Identification of drug-like phytochemicals in MMPs eases plant selection with one or more potential AOX-AINF-AAG-ASEN effects. To achieve the objectives, (I) we will create an MMP database through the literature examination to be analyzed for reported AOX-AINF-AAG-ASEN activity; (II) we will search for AOX-AINF-AAG-ASEN phytochemicals candidates in MMPs using chemoinformatics and network pharmacology; and (III) the selection process for superior plants involves the thorough assessment of their multiple activities and compounds while also considering their synergistic properties that interact with different action mechanisms, including AOX-AINF-AAG-ASEN actions.

## 2. Materials and Methods

### 2.1. Collection of Data

#### 2.1.1. Identification of MMPs

To delimit the size of the population of MMPs, a further review was performed up to June 2023. The list of MMPs was bought from diverse, credible sources, including the EncicloVida National Biodiversity Platform [20], Atlas de las Plantas de la Medicina Traditional Mexicana, and the Index and Synonymic of Mexican Medicinal Plants (IMEPLAN) [18].

#### 2.1.2. Search of AOX-AINF-AAG-ASEN Activity from MMPs

We reviewed bibliographies published in PubMed and Scopus up to July 2023 to find MMPs with or without scientific evidence related to AOX-AINF-AAG-ASEN activities.

#### 2.1.3. Bibliographic Review of the Phytochemicals Reported to MMPs

To obtain a complete dataset of MMPS-reported molecules, we searched PubMed, chemical abstracts, and Scopus until July in the year 2023.

#### 2.1.4. Creating MMP Dataset

We have created a dataset of MMPs, categorizing them according to family, genus, species, study type, synonymies, AOX-AINF-AAG-ASEN activities, other activities, diseases, and phytochemicals.

### 2.2. Identifying Plants with One or More AOX-AINF-AAG-ASEN Effects and Their Corresponding Phytochemicals

We analyzed the MMP dataset using scientific evidence to create a network that classified them by family, genus, and species and identified their AOX-AINF-AAG-ASEN capabilities. Using Cytoscape 3.8 and the Cytohubba plugin [27,28], the highly interconnected plants and their activity were identified.

### 2.3. MMPs Are Identified through Network Analysis Based on Taxonomic Features and AOX-AINF-AAG-ASEN Activities

We constructed a network derived from the MMP database based on family, genus, and species to locate plants without AOX-AINF-AAG-ASEN activity. These plants contain compounds with these reported activities. The network analysis was executed using the Cytoscape software version 3.8 [27]. The Cytohubba plugin [28] was employed to find the most connected plants and their reported activity in the network.

### 2.4. Collection of Drugs with AOX-AINF-AAG-ASEN Activity

A review of the approved drugs by the FDA in the Drug Bank (https://go.drugbank.com/) [29] accessed on 15 August 2022 concerning compounds reported as anti-oxidants and anti-inflammatories was performed. To create the reference dataset, we use senolytic and anti-aging compounds obtained from previous work [30,31].

#### Selection of Reference Compounds with One or More AOX-AINF-AAG-ASEN Activities through Hierarchical Analysis

The Simplified Molecule Input Line Entry System (SMILES) of the compounds with AOX-AINF-AAG-ASEN activity included in the drug dataset was examined in September 2022 in both PubChem [32] and in the SwissADME server (http://www.swissadme.ch accessed on 15 August 2022). We used Osiris Data Warrior V5.2.1 to analyze the physicochemical properties and molecular descriptors. The physicochemical properties and molecular descriptors determined in this work included G-protein-coupled receptors (GPCR) ligand (GPCR. Ligand), ion channel modulator (ion.channel.modulator), kinase inhibitor (kinase.inhibitor), nuclear receptor ligand (nuclear.receptor.ligand), protease inhibitor (protease.inhibitor), enzyme inhibitor (enzyme.inhibitor), number of violations (nviolations), number of atoms (natoms), log k p-values in cm/s (log.Kp..cm.s.), Lipinski’s Rule violations (Lipinski..violations), Ghose Filter violations (Ghose..violations), Veber Rule violations (Veber.violations), Egan Rule violations (Egan..violations), Muegge’s Rule violations (Muegge..violations), bioavailability score (bioavailability.score), molecular weight (molweight), P: conc (octanol)/conc (water) (cLogP), S: water solubility in mol/L (cLogS), hydrogen acceptor (H.acceptors), hydrogen donors (H.donors), total surface area (total.surface.area), polar surface area (polar.surface.area), drug likeness (drug likeness), shape index (shape.index), molecular flexibility (molecular.flexibility), electronegative atoms (electronegative.atoms), rotatable bonds (rotatable.bonds), aromatic rings (aromatic.rings), aromatic atoms (aromatic.atoms), sp3 atoms (sp3 atoms), and symmetric atoms (symmetric.atoms) already reported [30,33].

Using *K-means* and distance matrix, we clustered descriptors of AOX-AINF-AAG-ASEN compounds. For this analysis of clusters, we use the *complexheatmap* package [34] with R-Studio (R-Studio PBC250, Boston, MA, USA), version 3.4 [35].

### 2.5. Identifying Drug-like Phytochemicals from MMPs

#### Estimation of the Drug-likeness Index of Phytochemicals in the MMP Dataset Based on the Quantitative Estimate of Drug-Likeness (QED)

We were interested in detecting which phytochemicals from the literature had drug-like properties. To identify phytochemicals with favorable characteristics to oral administration. The accomplishment of the aim was facilitated by the application of DruLito’s drug-likeness estimator, a Quantitative Estimate of Drug-Likeness (QED) (http://www.niper.gov.in/pi_dev_tools/DruLiToWeb/DruLiTo_index.html accessed on 15 August 2022) [36]. The DruLito software (version 1.0.0) used SDF files from PubChem to estimate the drug likeness of NMMPs. We analyzed only those molecules available on this server.

We used QED to evaluate the drug likeness of the compounds. QED is a technique for evaluating drug likeness by concurrently considering the primary molecular characteristics. QED score is determined by molecular descriptors, such as molecular weight, Lap, HBA, HBD, rotatable bonds, TPSA, and aromatic bond count. QED goes from zero to one. Zero shows an unsuitable molecule, while one corresponds to molecules with favorable characteristics. We selected the phytochemicals with high QED scores (over 0.5).

### 2.6. Drug-like Phytochemicals Are Excluded Because of Toxicity and Lack of Targeting for AOX-AINF-AAG-ASEN

Drug-like phytochemicals were sieved according to the following two criteria: (1) exclusion of undesirable effects (namely, tumorigenic, reproductive, and irritant functions) and (2) preservation of AOX-AINF-AAG-ASEN compounds with target information accessible in the PubChem database (retrieved February 2023). The signaling pathways used to construct this network are cited as the inflammatory response pathway, the relationship between inflammation, COX and EGFR, interleukin-1 family signaling, TP53 regulates transcription of cell death genes, the intrinsic pathway of apoptosis, TNF-related weak inducer of apoptosis (TWEAK) signaling pathway, cell cycle, an overview of the interferons-mediated signaling pathway, DNA damage/telomere stress-induced senescence, oncogene-induced senescence, DNA damage response, oxidative stress-induced senescence, senescence-associated secretory phenotype (SASP), and apoptosis. Check the Appendix A for more information on plant data, targets, compounds, and network genes.

### 2.7. Identifying Drug-like Phytochemicals via Fingerprint Analysis with Reference Compounds

To determine their fingerprints and perform a comparison, we used *ChemmineR* and *rcd* in R-Studio version 3.4 [37]. We used the extended value with a default length of 1024 (number of bits), taking rings and atomic properties. We used AOX-AINF-ASEN compounds as references and phytochemicals as tested datasets. Then, we performed a cluster analysis using the Tanimoto coefficient to compare them. We used Ward’s clustering method to classify the molecules into three groups. To confirm the clustering results, the Dunn index and the silhouette coefficient were used [37]. We performed different clustering methods; the reference compound cluster was analyzed to obtain molecules with one or more potential AOX-AINF-AAG-ASEN activities.

### 2.8. Structural Network Analysis Identifies MMPs with One or More Potential AOX-AINF-AAG-ASEN Activities

A structural network was employed, which was constructed using drug-like phytochemicals having similarity to references and target genes related to inflammation, oxidative stress, and senescence pathways. This network was also built based on signaling pathways found in Homo sapiens, which are associated with inflammation, oxidative stress, senescence, and their related processes. The signaling pathways involved in these processes included the inflammatory pathway, the relationship between inflammation, COX and EGFR, interleukin-1 family signaling, TP53 regulates transcription of cell death genes, the intrinsic pathway of apoptosis, TNF-related weak inducer of apoptosis (TWEAK) signaling pathway, cell cycle, overview of interferons-mediated signaling pathway, DNA damage/telomere stress-induced senescence, oncogene-induced senescence, DNA damage response, oxidative stress-induced senescence, senescence-associated secretory phenotype (SASP), and apoptosis.

#### Enrichment Analysis of the Pharmacological Targets from the Most Relevant MMPs

To determine the most important pharmacological target, drug-like phytochemical pathways involved in the structural network, an analysis was performed. They were built using Cytoscape software v 3.8. The JEPETTO plugin of Cytoscape [38] was employed to identify the major signaling pathways involved in MMPs and that interact with their phytochemicals. The significantly enriched pathways (*p* < 0.05, *p*-values) were revised using the XD-score, agreeing with previous reports [39,40]. The KEGG database was used for this enrichment [41] (Appendix A contains the results from the enrichment analysis).

Highly connected genes were identified in the network using the Cytohubba plugin score.

### 2.9. Animal Care

The mice were obtained from the medicine school of UNAM with the support of project # 008-CIC-2022. Their care was in line with Mexican standards (NOM-062-ZOO-1999; NOM-087-SEMARNAT-SSA1-2002) [42,43], and international official guidelines (Guide for the Care and Use of Laboratory Animals, revised in 2011) [44].

#### 2.9.1. Experimental Testing

*L. guatemalensis* Benth (Leguminosae) was collected in Santiago Pinotepa Nacional Km 8 road Pinotepa Nacional-Jamiltepecsws, Oaxaca, Mexico. A voucher on the plant is registered in the Mexican National Herb (IBUNAM: MEXU: 1139179). For more details, see Appendix A. The dried root material was converted into chips and extracted with hexane for 72 h [45]. Male BALB/c 8-week-old mice (26.67 g) were separated randomly into six experimental groups of six mice each (*n* = 6). Each group received one of these pretreatments: indomethacin (Sigma-Aldrich Cat. I7378, St. Louis, MO, USA) 10 mg/kg, as previously reported in inflammation tests [46,47]. *L. guatemalensis* extract dissolved in corn oil and administered at 10, 31, and 100 mg/kg independently in each group, corn oil alone, and distilled water. *L. guatemalensis* doses were based on a chemotaxonomic study of a published article where the analgesic activity of another member of the *Lonchocarpus genus* (*Lonchocarpus sericeus*) was evaluated in mice [48]. The LD50 of the extract was used as a reference for the establishment of the rest of the doses [48]. The pretreatments were executed 30 min before the administration of carrageenan. In each group, inflammation was induced by administering a sub-plantar injection of 50 μL of 1% carrageenan (Sigma-Aldrich Cat. 22049, St. Louis, MO, USA) into the right paw, as earlier described by Winter et al. [49,50,51]. Simultaneously, in the left paw, distilled water was being administered. In the paw edema model, indomethacin, a potent non-steroidal anti-inflammatory drug (NSAID), is considered an excellent positive control due to its potent inhibition of vascular events [52]. The edema of the paw was determined using a digital micrometer produced by Mitutoyo. Measurements were conducted before carrageenan was introduced and at various time points after inoculation [53]. The edema rate was determined as per the previously specified method, employing the following equation: Edema (%) = (M_f_ − M_i_/M_i_) (100). Where M_f_ is the paw measurement at different times, and M_i_ is the paw measurement before inflammation was induced, both obtained from the same right paw [54].

#### 2.9.2. Statistical Analysis

The data are presented as the mean ± standard error of the mean (s.e.m) of the percentage of edema induced by the administration of carrageenan. Multiple group comparisons were performed using one-way analysis of variance (ANOVA) post hoc Dunnett’s test to detect inter-group differences. The difference was considered statistically significant when *p* < 0.05. The Graph Pad Prism 8.4.3 (GraphPad Software Inc., San Diego, CA, USA) software was used to perform the statistical analysis.

Figure 1 summarizes the methodology followed in this work.

## 3. Results

### 3.1. Identifying MMPs with and without Evidence of One or More AOX-AINF-AAG-ASEN Activities

Out of the 1025 MMPs that were consulted in EncicloVida [19], Atlas de las Plantas de la Medicina Traditional Mexicana, and Index and Synonymic of Mexican Medicinal Plants (IMEPLAN) [27], 364 demonstrated properties of AOX-AINF-AAG-ASEN, while 661 NMMPs lacked any such background. We list the results in Table 1.

The clustering and relationship between families, genera, and species of all MMPs and their activities of anti-oxidation, anti-inflammation, anti-aging, and anti-senescence are presented in Figure 2A. The analysis by the Cytohubba plugin showed that *Asteraceae*, *Fabaceae*, *Chromolaena*, *Malvaceae*, *Piperaceae*, and some genera like *Granulosa*, *Peperomia*, *Chromolaena*, *Mimosa*, and *Malvaviscus* (Figure 2B) have been extensively researched. Also, a large part of the MMPs lack any background related to AOX-AINF-AAG-ASEN activity.

#### 3.1.1. MMPs with AOX-AINF-AAG-ASEN Multi-Effect and Their Corresponding Phytochemicals

The Figure 2A network was employed with Cytohubba to identify successful multi-effect MMPS. The results are shown in Table 2. The most relevant families with multi-effects included *Dilleniaceae*, *Lauraceae*, *Olacaceae*, *Fabaceae*, *Arecaceae*, *Meliaceae*, *Asteraceae*, *Malvaceae*, *Solanaceae*, and *Sapotaceae*. *Ximenia americana*, *Curatella americana*, *Malvastrum coromandelianum*, and *Manilkara zapota* exhibited the most significant multi-effect activity because they fulfilled all the activities considered in this research. The plants under investigation display diverse chemical structures that are classified in multiple clusters, which were previously noted for their AOX-AINF-AAG-ASEN properties. The flavonoids present in these plants are predominantly active compounds.

#### 3.1.2. Identification of MMPs without Scientific Evidence of AOX-AINF-AAG-ASEN Properties

MMPs are shown in Figure 3A, categorized by family, genus, and species without AOX-AINF-AAG-ASEN. The phytochemical profile of the phytochemicals is included.

Figure 3B shows a network analysis considering those MMPs with a greater number of phytochemicals that showed activity. In Table 3 are listed the most relevant families of MMPs (*Apocynaceae*, *Fabaceae*, and *Lamiaceae*). Rutin, stigmasterol, apigenin, and oleanolic acid are the phytochemicals that display the highest level of interconnectedness and are also emphasized.

#### 3.1.3. Identification of Reference Compounds with Some Reported AOX-AINF-AAG-ASEN Activity through Hierarchical Analysis

Figure 4 shows a heatmap of the main molecular similarities found among compounds with some AOX-AINF-AAG-ASEN properties. We analyzed the chemical space using the *k-means* algorithm and hierarchical grouping. Three clusters of reference compounds with individual AOX-AINF-AAG-ASEN activity were identified through the organization of the molecular descriptors’ matrix. The top cluster in the heatmap (marked in a red rectangle) revealed the most similar compounds, which included homologous nucleoside antibiotics, macrolides, anti-oxidants, aminoglycosides, and cardiac glycosides. The analysis conducted allowed us to use these molecules as reference compounds for structural analysis, which was performed using fingerprints. Table 4 summarizes the chemical structure and pharmacological activity of some compounds from cluster one.

#### 3.1.4. Identification of Drug-like Phytochemicals from MMPs

The literature search yielded 1025 MMPs, and 1382 phytochemicals were obtained from the PubChem server in SDF format. The SDF format is crucial for analyzing the compounds in DruLito software (version 1.0.0) and obtaining molecular descriptors. This search has yielded a result indicating that only 1373 out of 1382 molecules are available in the SDF format. We used Quantitative Estimation of Drug-likeness (QED) to identify phytochemicals with drug-like properties, a criterion for early drug discovery. Table 5 shows the number of phytochemicals analyzed and the number that passed the rule with a score greater than 0.5. We obtained only 718 drug-like compounds.

#### 3.1.5. Drug-like Phytochemicals Were Excluded Because of Toxicity and No Target for AOX-AINF-AAG-ASEN Activity

Given the potential toxic effect of some phytochemicals, we conducted a toxicity risk assessment of our drug-like phytochemicals. We removed those with tumorigenic, irritant, or mutagenic properties, resulting in 417 compounds.

The 417 drug-like phytochemicals identified through DataWarrior were subjected to a supplementary filter. We analyzed the fingerprints of 148 drug-like phytochemicals with PubChem-reported targets.

#### 3.1.6. Identification of Drug-like Phytochemicals via Fingerprint Analysis with Reference Compounds

We performed a comparison between the reference compounds we obtained from the top cluster of Figure 4 and 148 drug-like phytochemicals from MMPs. The reference compounds include paromomycin, magnesium ascorbate, calcium ascorbate, rapamycin, digoxin, geldanamycin, alvespimicyn, oleuropein, strophanthin k, natamycin, streptomycin, oubain, tunicamycin, ginsenoside Rb1, rapamycin, roxithromycin, timosaponin a-III, and azithromycin. By a fingerprint analysis using the Tanimoto index, the similarity or disparity among compounds was determined. According to the molecular distances and Ward’s method, the molecules were classified into three groups by elbow method (Figure 5A). Figure 5B shows the similarity between phytochemicals and reference compounds in Ward’s cluster map. *K-means* (Figure 5C) and silhouette method (Figure 5D) corroborated these results. Paromomycin, magnesium ascorbate, calcium ascorbate, rapamycin, geldanamycin, alvespimicyn, oleuropein, natamycin, streptomycin, tunicamycin, rifaximin, and roxithromycin all have structural similarities with 60 drug-like phytochemicals from MMPs. A total of 50 phytochemicals with drug-like qualities share structural similarities with digoxin and strophanthin. Ouabain and ginsenoside share a similar structure with 49 drug-like phytochemicals (see Appendix A). Drug-like phytochemicals from MMPs show structural similarities with several reference compounds. We depict an example of this analysis in Figure 5, using paromomycin as a reference compound. At the end of fingerprint analysis, we obtained 109 drug-like phytochemicals like reference compounds.

#### 3.1.7. Identification of MMPs with One or More Potential AOX-AINF-AAG-ASEN Activities through Structural Network Analysis

We only considered MMPs that met these requirements: (1) Phytochemicals with QED value over 0.5 for considered drug-like phytochemicals. (2) Phytochemicals showing structural resemblance to reference compounds. (3) Phytochemicals with target genes reported in bioassays. (4) Phytochemicals lacking AOX-AINF-AAG-ASEN activity. From 109 drug-like phytochemicals with structural similarity to the reference compounds, we obtained only 24 that met the aforementioned criteria.

The network pharmacology analysis showed 2115 nodes and 4101 edges (Figure 6A). Four candidate MMPs with AOX-AINF-AAG-ASEN potential were obtained at the end of the analysis, namely *Lonchocarpus guatemalensis*, *Vallesia glabra*, *Erythrina oaxacana*, and *Erythrina sousae*, as shown in Figure 6B. Their respective family, genus, species, reported activity, part of the plant used in the studies, number of compounds with reported targets, and number of targets that interact with the network are detailed in Table 6. The phytochemicals that demonstrate the highest connectivity with networks associated with AOX-AINF-AAG-ASEN processes include apparicine, vallesine pseudo tropine, scopine, tropine, corytuberine, cryptocoryne, erysotrine, germanicol, laudanosine liriodenine, lonchocarpin, magnocurarine, and scopoline. Table 7 mentions their source, pharmacological activity, structural similarity, and potential activity.

#### 3.1.8. Enrichment Analysis of the Pharmacological Targets from the Most Relevant MMPs

To find several new associations, we determined the XD-score ranking. The most connected genes mentioned above (ABCB1, ALOX12, APOBEC3G, ATM, BRCA1, CHRNA4, CHRNB2, FEN1, MCOLN1, OTUD3, PINK1, POLI, POLK, PRKN, SMAD3, SMPD1, SNCA, TDP1, UCHL1, USP10, USP17L5, USP28, USP30, USP7, and USP8) were used to perform this analysis. The most relevant signaling pathways or processes associated with this pharmacological target gene related to these MMPs are shown in Appendix A.

#### 3.1.9. Effects of *L. guatemalensis* Benth on Carrageenan-Induced Paw Edema

To corroborate the results obtained in silico, and because of the accessibility of *Lonchocarpus guatemalensis*, an extract from the root of this plant was evaluated in a carrageenan-induced paw edema assay. The results can be seen in Figure 7. Carrageenan administration triggered a significant increase in the right paw volume of all pretreated groups. *L. guatemalensis* extract significantly inhibited the formation of the edema from 2 h up to 6 h at 31 and 100 mg/kg. The inhibition observed is comparable to that seen with the pre-treatment of indomethacin (10 mg/kg). The group treated with *L. guatemalensis* (10 mg/kg) did not show any statistically significant decrease in edema. The groups administered with vehicles, corn oil, or distilled water did not exhibit any impact on the rise of edema in the right paw.

Appendix A provide information on the determination of the percentage of anti-inflammatory effects caused by the hexanic extract of *L. guatemalensis.*

## 4. Discussion

The present study should rescue the knowledge about the truly native Mexican plants used medicinally. Motivating results show that only 35.51% of CONABIO’s 1025 MMPs underwent any study of AOX-AINF-AAG-ASEN activity, and 64.48% lack scientific reports on it. To recognize MMPs with potential AOX-AINF-AAG-ASEN phytochemicals through network analysis and chemoinformatics, a six-phase workflow was devised. This method enhances the probability of desired biological activity when tested.

Phase I. In MMPs, the anti-oxidant and anti-inflammatory activities have been the most studied (414 plants). The interest in these properties is consistent with the literature, which shows that various plant compounds possess helpful properties such as anti-oxidative, anti-inflammatory, and anticancer, among others [55]. MMPs such as *Ximenia Americana*, *Manilkara zapota*, and *Curatella Americana* have reported all the studied activities [56,57,58,59,60,61,62]. In preclinical medicinal research, these plants show promise in combating age-related, degenerative, and infectious diseases caused by inflammation, oxidative stress, and cellular senescence. Our networks of the MMPs show relevant nodes within the Asteraceae and Fabaceae families. Edible flowers acquired popularity for their polyphenol content, which has anti-oxidants and anti-inflammatory properties. They include Asteraceae [63] and Fabaceae [64] families. The polyphenols are present in several natural products like tea, chocolate, fruits, and vegetables. This provides anti-oxidant, cytotoxic, anti-inflammatory, antihypertensive, and anti-diabetic benefits [65].

Phase 2. During the network analysis, among 661 (54.6%) MMPs that lacked AOX-AINF-AAG-ASEN activities, *Plumeria rubra*, *Lonchocarpus yucatanensis*, *and Salvia polystachya* were highlighted because they contained many phytochemicals with some reported AOX-AINF-AAG-ASEN effect [66,67,68]. Predominant compounds in these MMPs include polyphenols (quercetin, rutin, apigenin) and phytosterols (Beta-sitosterol, Stigmasterol). Also, vanillic acid, ferulic acid, apigenin, luteolin, rosmarinic acid, alkaloids, and terpenes, among others [69,70,71]. This finding provides us with greater success when testing the extracts of these plants in experimental models.

Phase 3. Despite having 1025 MMPs, we found a great redundancy in the phytochemicals reported in these plants. A total of 1382 phytochemicals were found. Only 1373 were available on servers. Characterizing the phytochemicals and their pharmacological properties required a preliminary step because of their heterogeneous nature. Drug-like capacity using the QED criteria allowed us to establish if a compound is suitable for oral administration in an in vivo model. From the 1373 phytochemicals collected from the MMPs, only 52.3% (718 MMPs) met the requirements to be considered “drug-like compounds”. Of these drug-like compounds, only 58.07% (417) were free of tumorigenic, irritant, or mutagenic properties. The importance of these findings lies because some compounds may be harmful and not well absorbed when taken orally. These issues encourage the optimization of the discarded molecules to find new potential drugs [72] or use its functional fragments for the design of evolutionary libraries for the design of drugs in the future [73].

Selecting drug-like phytochemicals with bioassay targets improved the chances of finding potential activity molecules. Of 417 drug-like phytochemicals, only 148 (35%) had reported targets. The selection based on the target reported excluded potential candidates with structural similarity. These excluded phytochemicals may be considered if we use predictive techniques, such as molecular docking. However, under our focus, we use the network with predicted target c bioassays to obtain more targets experimentally and thus guarantee a better selection of compounds with potential activity by network pharmacology or other methodologies. Using predictive data to choose candidates for structural networks may create biases instead of using experimental bioassays. These issues would generate unsuccessful candidates when evaluated in an experimental model. For this reason, it is of great importance to conduct additional bioassays to obtain more experimental targets and thus ensure a better selection of compounds with potential activity through the use of network pharmacology or other methodologies.

Phase 4. For structural similarity analysis, we build a dataset with reference compounds. To achieve this aim, we use Drug Bank Online, a comprehensive and cost-free online resource that comprises intricate data on medication. The included data involve drug targets, drug actions, and the interactions between FDA-approved medicines, besides experimental drugs awaiting FDA approval. Drug Bank’s high-quality content and primary origin make it one of the world’s most used reference drug resources [56]. Reference molecules classified as steroids (6 molecules), macrolides (4 molecules), and macrocyclic compounds (4 molecules) are in Figure 4’s top cluster. Most reference compounds have chemotherapeutic activity against bacteria, parasites, and cancer cells. Some of them are used to treat cardiovascular illnesses (3 molecules), and one compound is an immunosuppressor (Drugbank online). Most of these compounds are contained in the common functional lactonic group (63.15%). Lactones are outstanding exponents of secondary metabolites because of their remarkable biological activities and chemical architectures. δ-Lactones are the most abundant because of the high stability of their lactone rings δ-lactone moiety is common in several natural compounds with diverse biological activities, such as HIV protease inhibition, apoptosis induction (goniothalamin and rasfonin), antileukemic (dictyopyrone C), antitumor (pironetin), leishmanicidal, trypanocidal, antifungal (argentilactone), antibacterial, anti-inflammatory, and anticancer. This characteristic makes them highly valuable reference compounds for our research [74]. The reference compounds that remain are used to treat cardiovascular illnesses. Various structures analyzed showed steroidal or steroid-related structures, including precursor steroids like triterpene [75]. Antimicrobial, anticancer anti-oxidant, immunomodulatory, and anti-inflammatory effects [74,76] are related to their steroidal structure. This characteristic makes them highly valuable reference compounds for our research.

Hierarchical and *K-means* clustering techniques were used in fingerprint analysis due to their success in previous research [30,31,37]. Our previous works were used for the search for senolytics [30,31]. We applied it in this investigation to phytochemicals using compounds with different pharmacological functions. The verification of the clustering analysis was performed by various means, including the Ward method, Dunn index, and silhouette coefficient, all of which confirmed the existence of the same drug-like phytochemicals when compared to the corresponding reference drug. Of 148 phytochemicals obtained from MMPS, 109 showed structural similarity with reference compounds.

Phase 5. An examination of the network’s nodes resulted in the identification of 24 compounds from a pool of 119 phytochemicals. It is noteworthy that most of these compounds are classified as alkaloids, among which are apparicine, vallesine, pseudo tropine, scopine, tropine, corytuberine, cryptocoryne, erysotrine, germanicol, laudanosine, liriodenine, magnocurarine, and scopoline. One criterion for selecting the phytochemicals is the similarity to reference compounds. Despite the predominant use of steroids, macrolides, and macrocyclic compounds as reference materials in this stage of the study, the structural analysis using fingerprints revealed that the phytochemicals with drug-like properties were mainly of an alkaloid nature. This trend may be because there is a greater representation of alkaloids, according to studies searching for phytochemicals in plants [77]. Within the human or animal body, alkaloids are responsible for dynamic biological activities. Research findings provide evidence of the benefits of alkaloids, which could be applied in discovering and designing new analogs that could be of therapeutic use in various treatments [78]. Alkaloids can provide feasible lead compounds for the discovery and development of anti-inflammatory drugs. Most of the alkaloids are simple in structure and easy to modify, providing a viable framework and pharmacophore options for the design of new anti-inflammatory drugs [79]. Another criterion for the selection of phytochemicals was the target identification. With the advancement of synthetic, computational, and biological approaches, structural simplification will play an increasingly important role in drug discovery and contribute to improving the efficiency and success rate of drug development [80]. Through the utilization of network analysis and chemoinformatics, *Lonchocarpus guatemalensis*, *Vallesia glabra*, *Erythrina oaxacana*, and *Erythrina sousae* have been identified as potential subjects for forthcoming pharmacological studies. The identification of candidate phytochemicals, including lonchocarpin, vallesine, and erysotrine, in these MMPs presents an intriguing opportunity for future chemotaxonomic research to examine their activity in several species of plants.

Our search for plants with potential drug-like phytochemicals was limited because of the exclusion of MMPs based on their reported active compounds or toxic effects.

Phase 6. The evidence shows that many drugs act via the modulation of multiple targets rather than a single target [81]. For this reason, the enrichment analysis was performed because of the limited number of targets of lonchocarpin, vallesine, and erysotrine found in *Lonchocarpus guatemalensis*, *Vallesia glabra*, *Erythrina oaxacana*, and *Erythrina sousae*. In our enrichment analysis, the targets identified are found in several cellular processes, infections, cancer, and age-related diseases, where oxidation, inflammation, senescence, and aging play an important role in their development. Four Mexican plants with medicinal potential for AOX-AINF-AAG-ASEN activities were identified through virtual screening and network analysis.

The application of these computational tools has played a pivotal role in the discovery and design of highly efficacious medications. The article highlights the crucial role that cheminformatics and network pharmacology play in the process of drug discovery. The fusion of computational techniques and extensive data analysis has brought about a paradigm shift in the selection of medicinal plants. This integration enables more efficient exploration of the chemical space and accurate prediction of both therapeutic and toxicological biological activities of phytochemicals. Our search methodology significantly highlights the essential characteristics necessary for the continued search and selection of medicinal plants based on phytochemicals. This approach transcends the boundaries of chemotaxonomic or ethnomedicinal studies, enabling the identification of potential candidates. Our model demonstrates its effectiveness in predicting molecular properties and activities, helping in the prioritization and optimization of phytochemicals.

Furthermore, clustering is one of the simplest and most popular unsupervised machine learning algorithms. *K-means*, supported by network pharmacology and fingerprint-based structural similarity methods, opens possibilities for computer-aided design and discovery of new molecules with the desired properties.

This convergence has great potential to speed up the drug discovery process, reduce costs, and increase success rates in the identification of new therapeutic agents. Continued research and development in this area of phytopharmacology will undoubtedly pave the way for more efficient and precise drug design strategies, ultimately benefiting patients and advancing the field of pharmaceutical sciences.

Phase 7. Of all the candidates, we selected *L. guatemalensis* because of its availability. The rest of the plants, such as *Vallesia glabra*, *Erythina oaxacana*, and *Erythina souzae,* are not in the flowering process. This study provides evidence that supports the anti-inflammatory effects of the hexane extract obtained from *L. guatemalensis*, specifically highlighting lonchocarpin as a significant constituent according to the existing literature. These results obtained on carrageenan-induced paw edema are preliminary, and further studies on the anti-inflammatory mechanisms of *L. guatemalensis* extracts, such as inflammation markers (determined in serum concentration, gene expression, etc.), are required. However, they are encouraged to be tested in other anti-oxidant or senescence models. This result should be noted that lonchocarpin demonstrates additional biological effects, such as its anti-oxidant and anticancer properties [82], which impact various targets involved in senescence processes [83]. All these aspects reinforce our strategy to select plants with several therapeutic effects, proposing *L guatemalensis* as a potential phyto-drug to treat different pathologies. Inflammation is the most common feature of many chronic diseases and complications while playing a critical role in carcinogenesis [84]. Several studies have shown that Nrf2 contributes to the early steps of the anti-inflammatory process by orchestrating the recruitment of inflammatory cells and regulating gene expression through the anti-oxidant response element (ARE). Therefore, lonchocarpin may be a potential agent for treating disorders that are associated with inflammation because it increases the expression of anti-oxidant enzymes, such as heme oxygenase-1 (HO-1), NAD(P)H: quinone oxidoreductase 1 (NQO1), and manganese superoxide dismutase (MnSOD), which are all under the control of Nrf2/anti-oxidant response element (ARE) signaling. Moreover, this phytochemical induces the nuclear translocation and DNA binding of Nrf2 to ARE as well as ARE-mediated transcriptional activities [85]. Also, lonchocarpin inhibits Wnt/β-catenin signaling and suppresses colorectal cancer proliferation. The inflammatory response must be tightly regulated because uncontrolled inflammation may lead to tissue injury. Among the many signaling pathways activated, the canonical Wnt/β-catenin has been recently shown to play an important role in the expression of several inflammatory molecules [86]. Also, Wnt/β-catenin mediates transcription and regulates oncogenic and immune responses [87] via the interaction with NF-κB, which is a key master of inflammation. Due to this reason, Wnt/β-catenin is a promising target for drug development [88]. *L. guatemalensis* induces inhibition of inflammation at early times, reducing the formation of edema induced by carrageenan simply due to the biological properties of lonchocarpin and, in particular, the inhibition of Wnt/β-catenin [82] gives us the probability that this chalcone is responsible for the effect of this plant [89], just as its presence has been related to this activity in other plants [90]. However, it is worth mentioning that other phytochemicals may also be participating in the anti-inflammatory activity.

The pharmaceutical industry has been confronting amplified drug development costs, intensified failure rates, augmented competition for established targets, and an escalated demand for first-in-class drug development predicated on new targets and pharmacological mechanisms [91]. We hold the belief that exploring medicinal plants and their phytochemicals is crucial for drug development. For this research, the information collected was analyzed by introducing biological big data analysis-based systems and chemoinformatics. Owing to the complexity of the phytochemical mixtures existing in plants, a multi-component and multi-target approach is required, which differs from the approach used with conventional medicines [92]. Qualitative and quantitative analysis of diverse phytochemicals and their association with systems biology is a crucial factor in the development of drugs from natural sources.

To know the poly-pharmacological effects of multiple components of medicinal plants, much time and effort are required through conventional experiments. It is, therefore, necessary to resort to integrative strategies to select and develop better treatments with lower costs. Efforts have been made to understand the disease by integrating conventional molecular biology experiments, bio-big data analysis, and artificial intelligence technologies, which are increasing rapidly [93]. Future research should optimize phytochemical analysis and network pharmacology to predict multiple targets. Combinatorial strategies that target multiple mechanisms may offer better chances of achieving clinically meaningful treatment [94]. Through implementing this integrative approach, we have a firm belief that our MMPs, chosen based on their phytochemical and pharmacological network components, are well-suitable candidates. The findings from this in vivo investigation of *Lonchocarpus guatemalensis* have confirmed the predicted anti-inflammatory activity observed in the in silico analysis, demonstrating promising results. However, additional in vitro studies with *Lonchocarpus guatemalensis* and experimental evaluation of other MMP candidates are necessary. We are currently conducting experimentation on these candidates, and we expect the results will be positive.

## 5. Conclusions

We discovered unexplored indigenous MMPs with fingerprint analysis and compound-target networks. The MMPs comprise phytochemicals that exhibit a similar structure to reference compounds. Their association with pharmacological bioassays shows their capability to elicit AOX-AINF-AAG-ASEN effects.

The following MMPs have been identified as the most successful because of their multi-effect activity: *Curatella americana*, *Persea americana*, *Ximenia americana*, *Parkinsonia aculeata*, *Acrocomia aculeata*, *Cedrela odorata*, *Chromolaena odorata*, *Malvastrum coromandelianum*, *Solanum nigrum*, and *Manilkara zapota*. Due to their phytochemical profiles and previously shown AOX-AINF-AAG-ASEN properties, *Plumeria rubra*, *Lonchocarpus yucatanensis*, and *Salvia polystachya* are promising sources of AOX-AINF-AAG-ASEN drug phytochemicals. *Lonchocarpus guatemalensis*, *Vallesia glabra*, *Erythrina oaxacana*, and *Erythrina sousae* because of their content of lonchocarpin, vallesine, and erysotrine, respectively, are the best candidates to show some AOX-AINF-AAG-ASEN effect. The experimental evaluation of *Lonchocarpus guatemalensis* Benth. in mice showed anti-inflammatory activity. This experimental evidence further confirmed our research strategy, marking the beginning of further study of other promising MMP candidates for the design of preclinical and subsequent clinical trials to test the efficacy and safety of these medicinal plants.

## Figures and Tables

**Figure 1 biomolecules-13-01673-f001:**
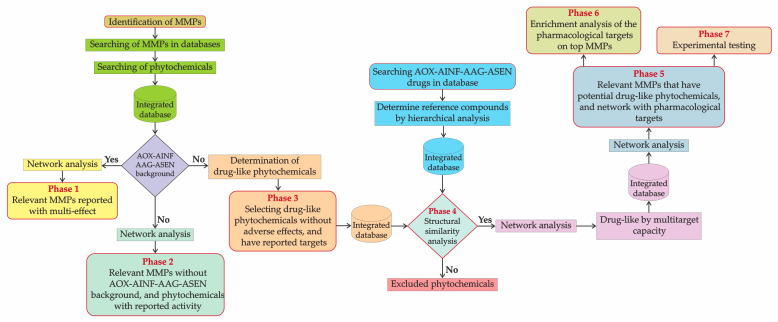
Workflow for identifying MMPs and drug-like phytochemicals related to AOX-AINF-AAG-ASEN activities. (1) Relevant MMPs by multi-effect reported. (2) Relevant MMPs without reported effect but with phytochemicals with reported activity. (3) Determination of drug-like phytochemicals from MMPs without AOX-AINF-AAG-ASEN background. (4) Structural comparison between drug-like phytochemicals and reference AOX-AINF-AAG-ASEN compounds. (5) MMPs that have potential drug-like phytochemicals can network with pharmacological targets. (6) Enrichment analysis of the pharmacological targets from the most relevant MMPs.

**Figure 2 biomolecules-13-01673-f002:**
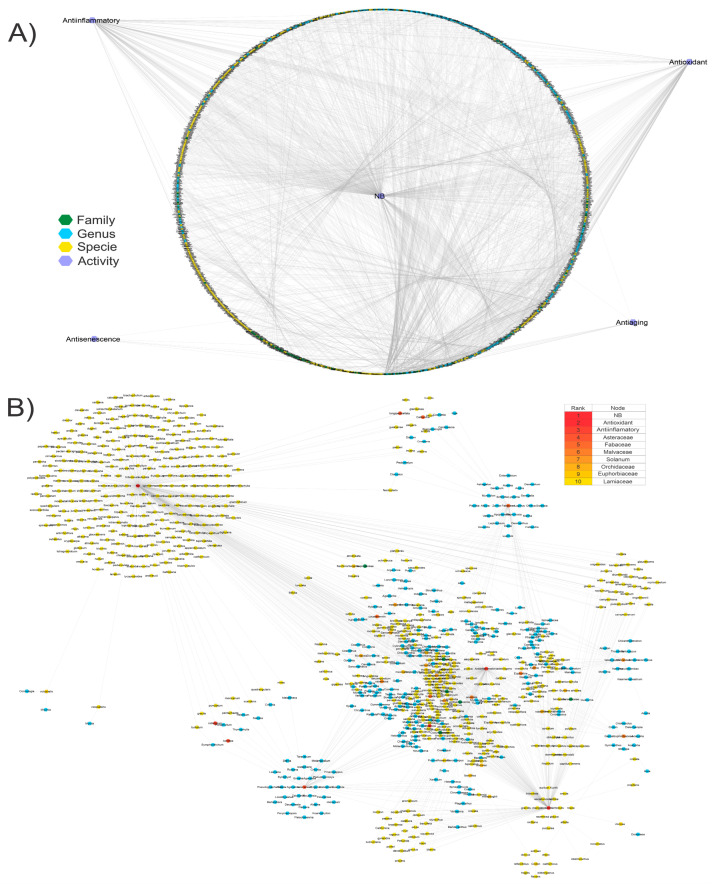
Graphical representation of the MMP dataset into a network. (**A**) The taxonomic characteristics of the MMPs are integrated into the structural network, which also includes information on the availability or lack of scientific support for AOX-AINF-AAG-ASEN properties. (**B**) Network with the most relevant node. The table summarizes the ten most connected nodes.

**Figure 3 biomolecules-13-01673-f003:**
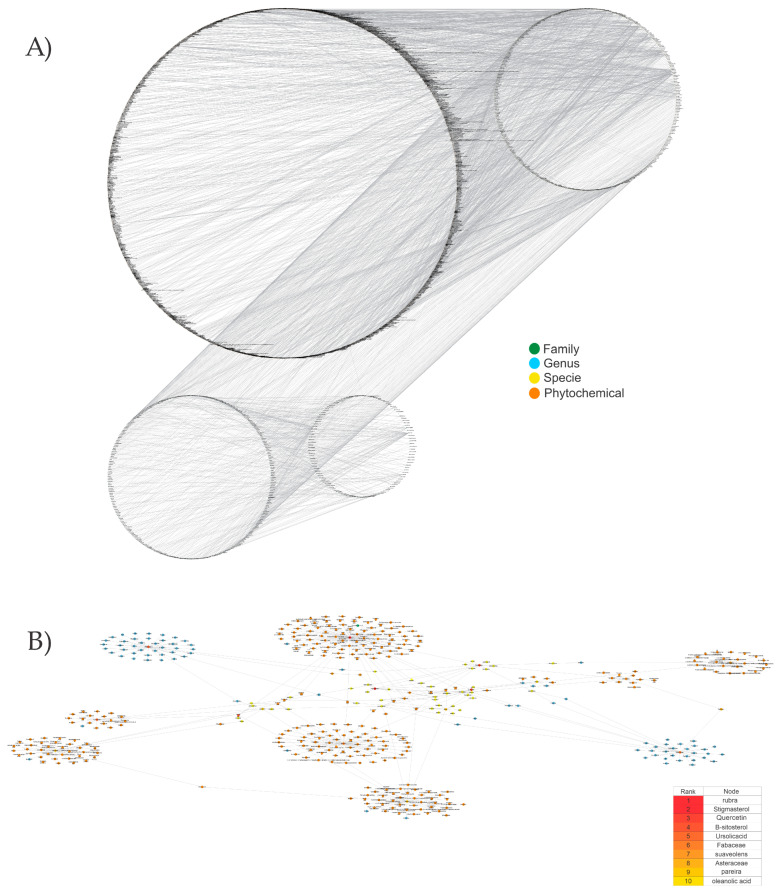
Structural network of MMPs that may have AOX-AINF-AAG-ASEN properties based on their chemical content. (**A**) MMP network lacks scientific evidence of AOX-AINF-AAG-ASEN activity but has proven phytochemicals. (**B**) The most relevant nodes of families, genera, and species of MMPs with potential AOX-AINF-AAG-ASEN activity and phytochemicals with reported studied activities. Table 3 shows the top 10 of the most interconnected nodes.

**Figure 4 biomolecules-13-01673-f004:**
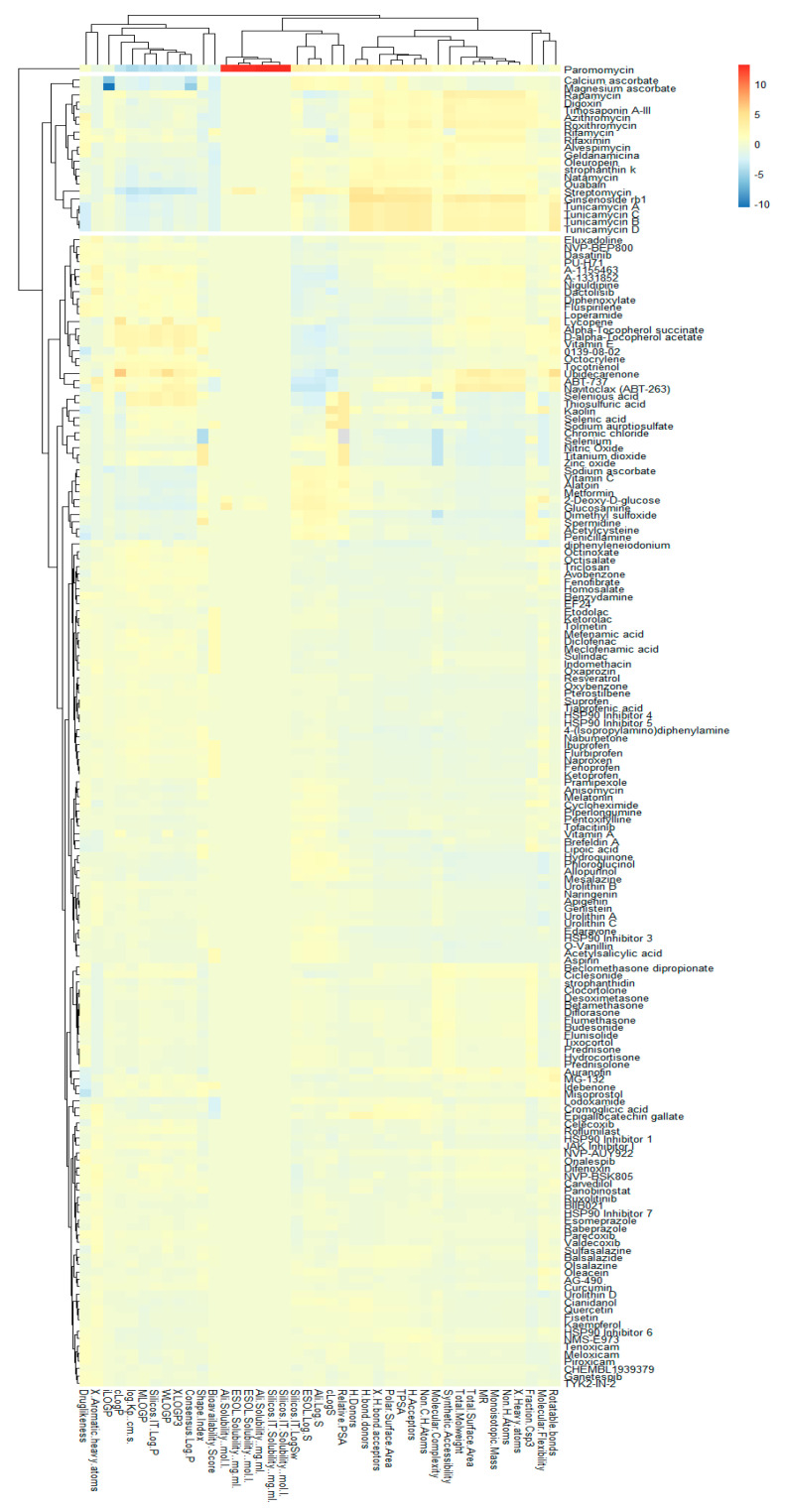
Hierarchical structural clustering of the AOX−AINF-AAG−ASEN compounds. The top cluster with the most similar compounds is shown and highlighted in a red rectangle. The cluster comprises the reference compounds paromomycin, magnesium ascorbate, calcium ascorbate, rapamycin, digoxin, geldanamycin, alvespimycin, oleuropein, strophanthin k, natamycin, streptomycin, ouabain, tunicamycin, ginsenoside Rb1, rapamycin, roxithromycin, timosaponin a-III, and azithromycin. It showed a similarity among molecular descriptors, such as cLogP, symmetric atoms, aromatic atoms, ring atoms, and log.kp.cm.

**Figure 5 biomolecules-13-01673-f005:**
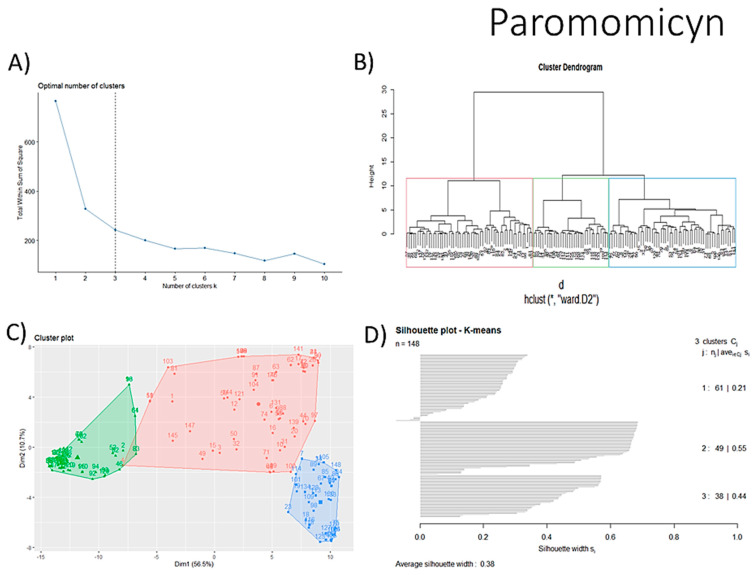
Steps of the process of selecting some AOX−AINF−AAG−ASEN potential compounds involved comparing 149 drug-like phytochemicals with the reference paromomycin based on their structural similarity. (**A**) The dashed line represents the suitable number of clusters determined by the Elbow method. (**B**) Cluster analysis of Ward’s method; the senolytic cluster is shaped in red and the senolytic molecule as 1. (**C**) Cluster plot using *K-means* showed the same molecules in the senolytic cluster marked in blue. (**D**) Silhouette cluster representation to corroborate the previously described cluster analysis methods. * Means matrix of distances used in clustering by Ward’s method in (**B**).

**Figure 6 biomolecules-13-01673-f006:**
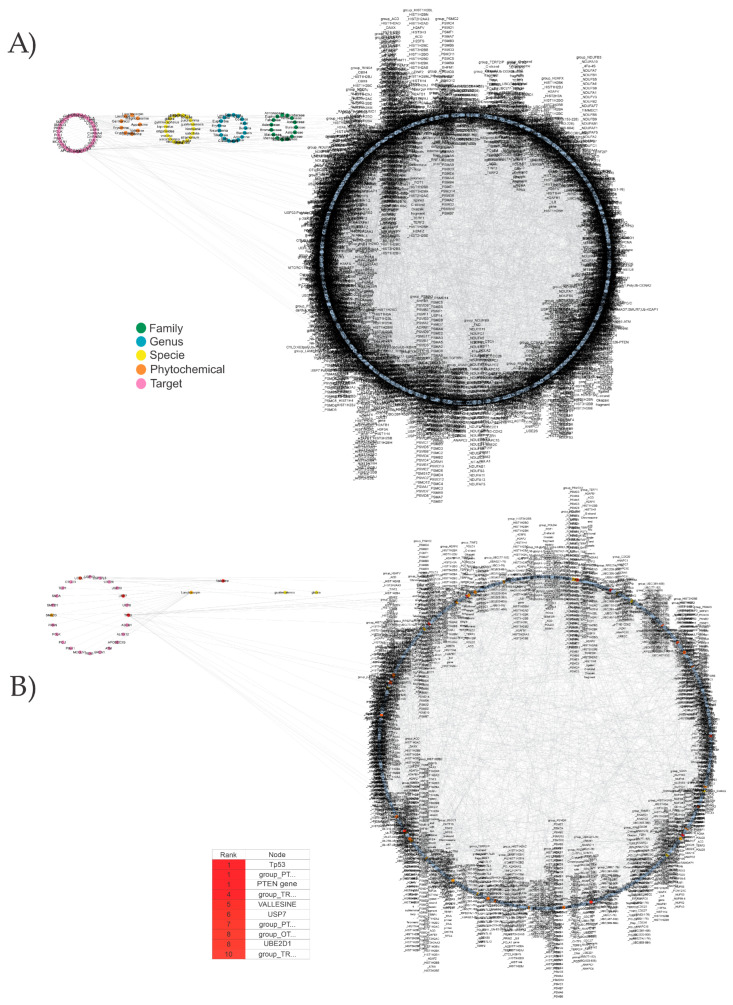
MMP network lacking AOX-AINF-AAG-ASEN properties with their phytochemicals and biological targets. (**A**) The network was constructed using MMPs, their putative AOX-AINF-AAG-ASEN compounds, and their biological targets reported in PubChem. The signaling pathways involved in these processes were many (see Methods). The network was designed and analyzed using Cytoscape software. (**B**) The structural network was built with the Cytohubba plugin for the top 10 most connected nodes.

**Figure 7 biomolecules-13-01673-f007:**
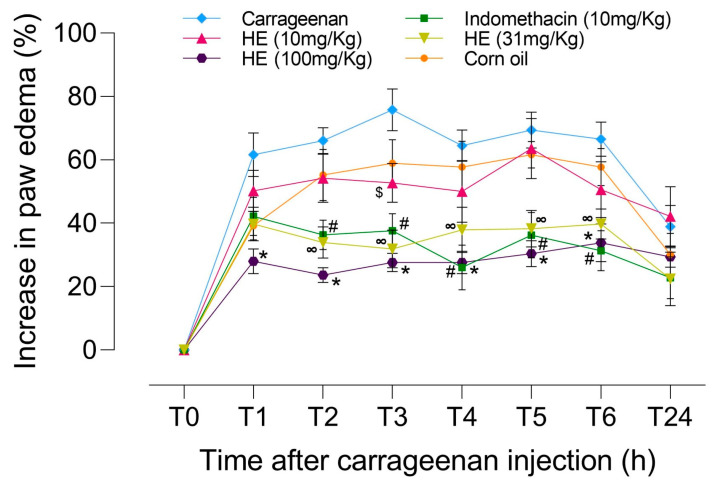
Effects of *Lonchocarpus guatemalensis* (10, 31, and 100 mg/kg) and indomethacin (10 mg/kg) on carrageenan-induced paw edema in mice. A statistically significant reduction in the percentage of edema can be observed from 1 to 6 h during treatment with *L. guatemalensis* at 30 and 100 mg/kg, which are similar to anti-inflammatory positive control indomethacin. Values are expressed as mean ± standard error of the mean (*n* = 6). Different symbols show significant (*p* < 0.05) differences between treatments according to ANOVA) post hoc Dunnett’s test. # carrageenan vs. indomethacin. $ carrageenan vs. HE (10 mg/kg). ∞ carrageenan vs. HE (31 mg/kg). * carrageenan vs. HE (100 mg/kg).

**Table 1 biomolecules-13-01673-t001:** The quantity of MMPs with or without evidence of some AOX-AINF-AAG-ASEN activity.

NMMPs Collected from the Literature	Number
Total	1025
With anti-oxidant activity	218
With anti-inflammatory activity	196
With anti-senescence activity	6
With anti-aging activity	14
Without AOX-AINF-AAG-ASEN activity	661

**Table 2 biomolecules-13-01673-t002:** The most relevant MMPs with reported multi-effect and their phytochemicals.

Family	Genus	Species	Phytochemical Profile	Reported Activities
Dilleniaceae	*Curatella*	*americana*	Avicularin, gallic acid, quercetin, quercetin-3-O-galactopyranoside, quercetin galactoarabinoside, quercetin-3-glucoside, quercetin-3-O-Alpha-l-rhamnoside, procyanidin, β-amyrin, betulinic acid, lupeol, and foeniculin	Anti-oxidant, anti-inflammatory, and anti-aging
Lauraceae	*Persea*	*americana*	Gallic acid, alfa-carotene, beta-carotene, vanillic acid, quercetin, catechin, epicatechin, procyanidins, tocopherols, lutein, glutathione, kaempferol, and chlorogenic acid	Anti-oxidant and anti-inflammatory
Olacaceae	*Ximenia*	*americana*	Quinic acid, coumaroyl-o-galloyl glucose, (epi)-catechin-(epi)-catechin-(epi)-catechin, (epi)-catechin carboxylic ester, (epi)-catechin-(epi)-catechin, procyanidin b1, epicatechin, catechin, pyrogallol-o-methylgalloyl glucose, procyanidin dimer monogallate, quercetin galloyl-hexoside, rutin, isoquercetin, avicularin, kaempferol 3-o-glucoside, kaempferol 3-neohesperidoside, quercetin pentoside, quercetin rhamnoside, kaempferol 3-o-arabinoside, kaempferol pentoside, uronic acid, arabinose, rhamnose, galactose, and glucose	Anti-oxidant, anti-inflammatory, anti-senescence, and anti-aging
Fabaceae	*Parkinsonia*	*aculeata*	4-hydroxyhematoxylol, apigenine, chrysoeriol, diosmetin beta-glucoside, kaempferol, lucenine 2, luteolin, orientin, iso-orientin, coumaric acid, rosmarinic acid, 7-O-Methyl-cyanidin-3-O-galactoside, kaempferol-8-C-β-D-glucoside, Luteolin-8-C-β-D-glucopyranoside, diosmetin-8-C-glucoside, apigenin-7-O-glucoside, kaempferol-3-O-glucoside, and homoplantaginin	Anti-oxidant and anti-inflammatory
Arecaceae	*Acrocomia*	*aculeata*	Gallic acid, caffeic acid, vanillic acid, rutin, quercetin, campesterol, stigmasterol, β-sitosterol, lupeol acetate, oleic acid, Docosahexaenoic acid, Eicosapentaenoic acid, and gamma-linolenic acid	Anti-oxidant and anti-inflammatory
Meliaceae	*Cedrela*	*odorata*	Kaempferol-3-O-β-D-glucopyranoside Kaempferide-3-O-β-D-rutinoside, and Kaempferide-3-O-β-D-rutinosyl-7-O-α-L-rhamnopyranoside	Anti-oxidant and anti-inflammatory
Asteraceae	*Chromolaena*	*odorata*	Caryophyllene, quercetin-4 methyl ether, aromadendrin-4′-methyl ether, taxifolin-7-methyl ether, taxifolin-4′-methyl ether, quercetin-7-methyl ether, kaempferol-4′-methyl ether, eridicytol-7, 4′-dimethyl ether, quercetin-7,4′-dimethyl ether, coriolic acid, coriolic acid methyl ester, 15-16-didehydrocoriolic acid, 15,16-didehydrocoriolic acid methyl ester, linoleamide, tamarixetin, kaempferide, 2-4-dihydroxy-3′,4′,6′-trimethoxychalcone, and 5,3′-dihydroxy-7,6′-dimethoxyflavanone or	Anti-oxidant and anti-inflammatory
Malvaceae	*Malvastrum*	*coromandelianum*	Palmitic acid, hexahydrofarnesyl acetone, linoleic acid, β-caryophyllene, cedren-8-en-15-ol, and arabinoxylan	Anti-oxidant, anti-inflammatory, anti-senescence, and anti-aging
Solanaceae	*Solanum*	*nigrum*	Acanthoside D, adenine, adenosine, allantoin, betaine, caffeic acid, cannabis in f, quercetin, quercitrin, stigmasterol, succinic acid, syringaresinol, tigogenin, tomatidenol, trans ferulic acid, trigonelline, ursolic acid, uttronin a, uttronin b, and vanillic acid	Anti-oxidant and anti-inflammatory
Sapotaceae	*Manilkara*	*zapota*	Taraxerol methyl ether, spinasterol, 6-hydroxy flavanone, dihydrokaempferol, 3,4-dihydroxy benzoic acid, taraxerol, taraxerone, lupeol acetate, epicatechin, gallocatechin, gallic acid, quercetin, myricitrin, catechin, myricetin, vanillic acid, caffeic acid, ferulic acid, syringic acid, afzelechin, and epigallocatechin	Anti-oxidant, anti-inflammatory, anti-senescence, and anti-aging

**Table 3 biomolecules-13-01673-t003:** The most relevant potential MMPs without reported AOX-AINF-AAG-ASEN effect and their phytochemicals.

Family	Genus	Species	Score
Apocynaceae	*Plumeria*	*rubra*	1023
Fabaceae	*Lonchocarpus*	*yucatanensis*	187
Lamiaceae	*Salvia*	*polystachya*	150
**Phytochemicals**			**Score**
Beta-sitosterol			329
Quercetin			302
Ursolic acid			288
Rutin			197
Stigmasterol			196
Apigenin			101
Oleanolic acid			93
Thymol			51
Caffeic acid			51
Gallic acid			51

**Table 4 biomolecules-13-01673-t004:** The structure and pharmacological activity of the molecules from the top cluster, which possess some AOX-AINF-AAG-ASEN properties, are presented.

Compound	Structure	Predominance Pharmacological Activity *
Paromomycin	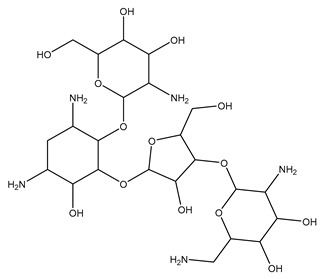	Antidiarrheal, antibiotic, and intestinal anti-inflammatory/anti-infective agent
Magnesium Ascorbate	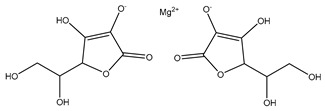	Anti-oxidant agent
Tunicamycin A-D	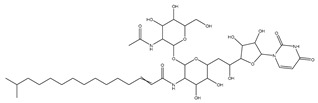	Antibiotic agent
Ginsenoside Rb1	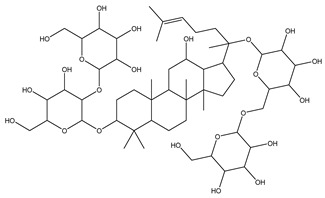	Anti-diabetic agent
Calcium Ascorbate	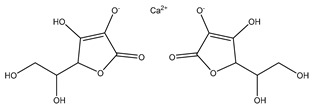	Anti-oxidant agent
Roxithromycin	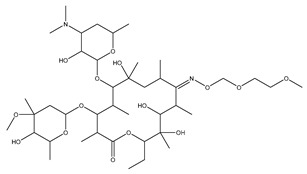	Antibiotic agent
Timosaponin a-III	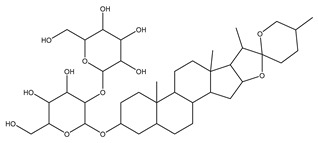	Anticancer and anti-inflammatory agent
Digoxin	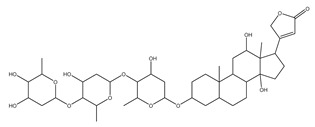	Positive inotropic and negative chronotropic agent
Rapamycin	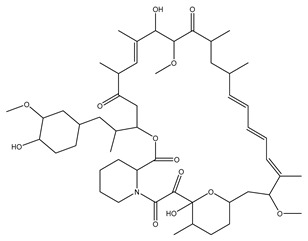	Anticancer agent

* Pharmacological activity was obtained by a literature review. The reference compounds had pharmacological effects on inflammation, oxidative stress, aging, and senescence.

**Table 5 biomolecules-13-01673-t005:** Several drug-like molecules from the total of phytochemicals obtained from MMPs.

	Number of Molecules from MMPs	Number of Molecules with a Score of Less than 0.5	Number of Molecules with a Score Greater than 0.5
QED	1373	605	718

**Table 6 biomolecules-13-01673-t006:** Relevant MPP candidates lack AOX-AINF-AAG-ASEN effects and have pharmacological targets.

Family	Genus	Species	Reported Activity *	Part of the Plant Used	Number of Compounds with Reported Targets	Number of Targets That Interact with the Network
Fabaceae	*Lonchocarpus*	*guatemalensis*	Antiacid activity	Roots	1	*2*
Apocynaceae	*Vallesia*	*glabra*	ND	Leaves, fruits, seeds, and milk-clotting	1	*8*
Fabaceae	*Erythrina*	*oaxacana*	ND	Flowers and seeds	1	*2*
Fabaceae	*Erythrina*	*sousae*	ND	Flowers and seeds	1	*2*

* The pharmacological activity was obtained by a literature review (for more details, see Appendix A).

**Table 7 biomolecules-13-01673-t007:** Relevant phytochemical candidates lacking AOX-AINF-AAG-ASEN effects and with pharmacological targets.

Phytochemicals	CID	Source MMPs	Pharmacological Activity	Structural Similarity with	Potential Activity
Apparicine	5281349	*Vallesia glabra*	Cytotoxic activity	Paromomycin, calcium ascorbate, magnesium ascorbate, rapamycin, timosaponin III, tunicamycin, geldanamycin, streptomycin, rifaximin, rifampicin, alvespimicyn, azithromycin, natamycin, roxithromycin, and oleuropein	Anti-oxidant, anti-inflammatory, and senolytic
Vallesine	20054841	*Vallesia glabra*	Antimalarial, diuretic, and respiratory stimulant activity	Paromomycin, calcium ascorbate, magnesium ascorbate, rapamycin, timosaponin III, tunicamycin, geldanamycin, streptomycin, rifaximin, rifampicin, alvespimicyn, azithromycin, natamycin, roxithromycin, and oleuropein	Anti-oxidant, anti-inflammatory, and analytic
Pseudotropine	449293	*Datura stramonium*	ND	Digoxin and strophanthin k	Senolytic
Scopine	1274465	*Datura stramonium*	α1-adrenergic receptor agonist	Paromomycin, calcium ascorbate, magnesium ascorbate, rapamycin, timosaponin III, tunicamycin, geldanamycin, streptomycin, rifaximin, rifampicin, alvespimicyn, azithromycin, natamycin, roxithromycin, and oleuropein	Anti-oxidant, anti-inflammatory, and senolytic
Tropine	8424	*Datura stramonium*	Acetylcholine antagonist	Oubain and ginsenoside Rb1	Senolytic
Corytuberine	160500	*Cissampelos pareira*	Inhibitor activity against malonyl-CoA: acyl carrier protein transacylase (MCAT) from Helicobacter pylori	Paromomycin, calcium ascorbate, magnesium ascorbate, rapamycin, timosaponin III, tunicamycin, geldanamycin, streptomycin, rifaximin, rifampicin, alvespimicyn, azithromycin, natamycin, roxithromycin, and oleuropein	Anti-oxidant, anti-inflammatory, and senolytic
Cryptocavine	72616	*Argemone platyceras*	Antibacterial activity	Paromomycin, calcium ascorbate, magnesium ascorbate, rapamycin, timosaponin III, tunicamycin, geldanamycin, streptomycin, rifaximin, rifampicin, alvespimicyn, azithromycin, natamycin, roxithromycin, and oleuropein	Anti-oxidant, anti-inflammatory, and senolytic
Erysotrine	442219	*Erythrina oaxacana* and *Erythrina sousae*	Cytotoxic and anticancer activity	Paromomycin, calcium ascorbate, magnesium ascorbate, rapamycin, timosaponin III, tunicamycin, geldanamycin, streptomycin, rifaximin, rifampicin, alvespimicyn, azithromycin, natamycin, roxithromycin, and oleuropein	Anti-oxidant, anti-inflammatory, and senolytic
Germanicol	122857	*Euphorbia pulcherrima*	Antiproliferative activity	Oubain and ginsenoside Rb1	
Laudanosine	15548	*Cissampelos pareira* and *Argemone platyceras*	Neuromuscular blocking activity	Paromomycin, calcium ascorbate, magnesium ascorbate, rapamycin, timosaponin III, tunicamycin, geldanamycin, streptomycin, rifaximin, rifampicin, alvespimicyn, azithromycin, natamycin, roxithromycin, and oleuropein	Anti-oxidant, anti-inflammatory, and senolytic
Liriodenine	10144	*Annona reticulata*	Antibacterial and antifungal activity	Paromomycin, calcium ascorbate, magnesium ascorbate, rapamycin, timosaponin III, tunicamycin, geldanamycin, streptomycin, rifaximin, rifampicin, alvespimicyn, azithromycin, natamycin, roxithromycin, and oleuropein	Anti-oxidant, anti-inflammatory, and senolytic
Lonchocarpin	6283743	*Lonchocarpus guatemalensis*	Inhibitor of the Wnt/β-catenin pathway	Paromomycin, calcium ascorbate, magnesium ascorbate, rapamycin, timosaponin III, tunicamycin, geldanamycin, streptomycin, rifaximin, and rifampicin	Anti-oxidant, anti-inflammatory, and senolytic
Magnocurarine	53266	*Cissampelos pareira*	ND	Paromomycin, calcium ascorbate, magnesium ascorbate, rapamycin, timosaponin III, tunicamycin, geldanamycin, streptomycin, rifaximin, and rifampicin	Anti-oxidant, anti-inflammatory, and senolytic
Scopoline	261184	*Datura stramonium*	Anticholinergic activity	Paromomycin, calcium ascorbate, magnesium ascorbate, rapamycin, timosaponin III, tunicamycin, geldanamycin, streptomycin, rifaximin, rifampicin, alvespimicyn, azithromycin, natamycin, roxithromycin, and oleuropein	Anti-oxidant, anti-inflammatory, and senolytic

## Data Availability

The data presented in this study are available in the Appendix A.

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
