# Peer review of "Selection of Mexican Medicinal Plants by Identification of Potential Phytochemicals with Anti-Aging, Anti-Inflammatory, and Anti-Oxidant Properties through Network Analysis and Chemoinformatic Screening"

_biomolecules, 2023, doi:10.3390/biom13111673_

Round 1
Reviewer 1 Report
Comments and Suggestions for Authors
The topic covered by the manuscript entitled "Selection of Mexican medicinal plants by identification of potential phytochemicals with anti-aging, anti-inflammatory, and antioxidant properties through network analysis and chemoinformatic screening: A mixed study review" covers important areas for the medical field.
Overall, the paper could bring valuable information, but the approach should be improved
The main positive aspects are the network analysis and chemoinformatic screening, still, the quality of the images included for this section results should be improved (all the figures)
The text is not very scientific (consider the ISI journal), especially in the case of the introduction - lines 39-52 have a too basic level, please modify these sentences
Basic is also the methodology used to prove the anti-inflammatory activity
If this is the only available method for the authors to use, then the conclusion could be slightly rephrased (the text appears to be already modified), but I personally do not find "validated ..." the best option
The most complex part of the study is represented by the network analysis and chemoinformatic screening - these results should be better highlighted, while the carrageenan-induced paw edema test is a literature-cited test, still its value is too narrow to validate
There are other more complex methods to evaluate anti-inflammatory potential and mechanisms based on the inflammation markers (serum concentration, gene expression etc)
the authors could mention that also the results obtained by the carrageenan-induced paw edema test are encouraging and that further studies are intended ...
replace validated with "further confirmed" or in vivo confirmed
Comments on the Quality of English Language
Minor editing of English language required
Author Response
Dear reviewer:
On behalf of the authors, we thank you for your suggestions and observations. Please see the attachment

Reviewer 2 Report
Comments and Suggestions for Authors
- On which base choose the dose of indomethacin 10 mg/kg., L. guatemalensis 10, 31, and 100 mg/kg and 50 μL of 1% carrageenan ( please insert the referance)
- All abbrevation in abstract section should write in full name such as MMPs, AOX-AINF-AAG-ASEN
- The manuscript clear, relevant for the field and presented in a well-structured manner
- The references should be improved with recent referances
- The manuscript scientifically sound
Moderate editing of English language required
Author Response

(The authors gave the same response as above.)
